# Integrating Juggling with Math Lessons: A Randomized Controlled Trial Assessing Effects of Physically Active Learning on Maths Performance and Enjoyment in Primary School Children

**DOI:** 10.3390/ijerph16142452

**Published:** 2019-07-10

**Authors:** Vera van den Berg, Amika S. Singh, Annet Komen, Chris Hazelebach, Ivo van Hilvoorde, Mai J. M. Chinapaw

**Affiliations:** 1Department of Public and Occupational Health, Amsterdam UMC, Vrije Universiteit Amsterdam, Amsterdam Public Health, van der Boechorststraat 7, 1081 BT Amsterdam, The Netherlands; 2Department of Movement and Education, Windesheim University of Applied Sciences, 8017 CA Zwolle, The Netherlands; 3Faculty of Behavioural and movement sciences, Vrije Universiteit Amsterdam, van der Boechorststraat 7, 1081 BT Amsterdam, The Netherlands

**Keywords:** physical active learning, mathematics, academic performance, enjoyment, children

## Abstract

There are tentative indications that physical activity (PA) during school time can be beneficial for children’s academic performance. So far, most studies have focused on the effects of moderate-to-vigorous PA, for example, in the form of energizers or extra physical education lessons. Little is known about the effects of physically active learning, in which PA is integrated with the academic content of the lessons, especially in preadolescent children. Moreover, there is a lack of knowledge regarding the enjoyment of physically active learning in this age group. Therefore, the aim of the current study was to assess the effects of integrating juggling with math practice in primary school children, on (1) multiplication memorization performance and (2) enjoyment during the math lessons. We conducted a cluster randomized controlled trial, in which 312 children (mean age 10.4 years) from nine Dutch primary schools participated. Fourteen classes were randomly assigned to either a group that learned juggling whilst practicing multiplication tables (intervention group), or to a group that practiced the same multiplication tables while sedentary (control group). Both interventions had a duration of 5 weeks and consisted of 20 short lessons (4 lessons per week, 5 to 8 min). We used mixed-model analyses to examine the effect of the intervention on multiplication memorization performance. Group (control or intervention) was used as the fixed factor, and class and school as random intercepts. Analyses were adjusted for pretest multiplication performance, age, gender, general motor skill level, physical activity behavior (PAQ-C), and academic math performance. No significant intervention effect on multiplication performance were observed. However, the math-juggling program significantly increased enjoyment of children during the math lessons. We can conclude that the intervention did not improve, but neither deteriorated children’s math performance. The increased enjoyment in the math-juggling group can serve as an important starting point for structurally incorporating physical activities in the classroom setting.

## 1. Introduction

There are tentative indications that physical activity (PA) during school time can be beneficial for children’s cognitive and academic performance, both in the short- and long-term [1,2,3,4,5,6]. So far, most studies have focused on the effects of stand-alone moderate-to-vigorous PA, for example, in the form of activity breaks between lessons or extra physical education lessons. In these type of PA programs, the movements are unrelated to and not integrated with academic content [7].

Far less is known about the effects of physically active learning (PAL), i.e., when PA is integrated with academic content. Some studies have shown positive effects of PAL programs on academic performance [8,9,10,11,12,13,14,15,16], while others found no improvements [17,18,19] or only improvements for a selection of the academic outcomes assessed [20,21,22,23]. Inconsistencies in outcomes may partly be due to the fact that studies show large variations with regard to the degree of integration of PA with academic content. Most studies put the academic content central and combine it with a motor task, such as running or jumping, while working on academic tasks (e.g., see References [20,22]). Other studies combine learning with a motor task that is highly integrated with the academic task, such as moving like an animal when learning about the continents and animals that live there (e.g., see Reference [13]), or rope skipping with addition sums linked to forward skipping and subtraction sums to backward skipping (e.g., see Reference [16]). Also, there are considerable differences in the duration of implemented PAL programs (1 week to 3 years), frequency (once or twice daily; one to five days per week), and duration of the lessons (10 min to 60 min). Relatively little is known about the effects of PAL for preadolescent children as most studies have focused on 4-to-5 (e.g., see References [11,12,13,14,20,21]) or 7-to-9-years-old children (e.g., see References [8,9,16,17,19,22,23]).

Many studies evaluating PAL on academic performance compare the intervention group to a regular curriculum control group with a different academic content. Based on these studies, it is not possible to draw firm conclusions on the added value of combining PA with academic content as the academic content also differs between groups [1]. When assessing the effects of programs combining PA with academic content, it is therefore important to offer the same academic content to both groups, also taking into account the amount of time spent on academic content.

Several researchers have suggested that the complexity, and thus level, of cognitive engagement of the motor task may play an important role in the effectiveness of PA programs on cognitive and academic outcomes [24,25,26]. Recent meta-analyses reported larger effects of long-term cognitively engaging PA (e.g., coordinative PA, team games, or complex motor skill training that require cognitive effort) on academic performance compared to non-cognitively engaging PA (e.g., simple, repetitive aerobic exercises) in children [3,5]. Differential effects between exercise types may stem from various reasons. Variability of practice (novelty, diversity/variety, and effort) that is central to coordination and perceptual–motor exercise training, for example, may be an interface between motor and cognitive development promotion [24,27]. Furthermore, it has been suggested that effects of PA programs on cognitive outcomes can be influenced by the enjoyment linked to participation in the PA program [25,28], with varied and challenging PA as factors that determine how much fun children experience [29]. In a recent study, we reported on children’s perspectives on integrating more PA in the school curriculum [30]. One of the major points children raised was the fact that PA needs to be fun. As the vast majority of previous PAL studies have focused on the integration of academic content with relatively simple aerobic exercises, there is an unmistakable lack of knowledge on the effects of PAL in which academic content is combined with a (new) challenging, potentially enjoyable motor task on children’s academic performance.

Schools are dynamic, complex systems where the focus is on learning [31] and meeting rigorous requirements for academic achievement within a limited time and budget. Therefore, it is exceedingly difficult to implement school-based PA programs. Teachers struggle with implementing PA in the regular academic curriculum [32,33], with a lack of time as the foremost important reason [32,34]. They are also hesitant to replace (parts of) regular lessons with physical activity. The primary challenge for any school-based program is to achieve high levels of adherence and compliance from teachers when it comes to the implementation of PA programs in the classroom setting. Therefore, programs should fit teachers’ preferences, i.e., additional PA programs must not be at the cost of too much time for academic lessons and require little or no preparation time [34].

Integrating physical activity with academic learning (PAL) addresses this “tug-of-war” between competing priorities [35] and might, therefore, increase teachers’ motivation for the structural implementation of PA in the school setting. However, some primary school teachers have questioned how PAL can be applied in the curriculum of the highest grades of primary school, i.e., for 10- to 12-year-old children [34,36]. Therefore, research on PAL in this age group is warranted. Another important aspect of PAL that has received relatively little attention so far is the effect that PAL may have on children’s enjoyment. Children’s enjoyment of PAL may be a reason for teachers to structurally implement PA programs and could be a potential facilitating factor for its implementation [37]. Additionally, enjoyment of PAL may benefit children’s motivation for learning, which is a possible mechanism for enhancing their academic performance [38]. In this respect, some studies have shown that preschool children (4–5 years) enjoyed PAL significantly more than sedentary lessons in which the same academic content was provided [12,13,14]. To the best of our knowledge, it is still unknown how preadolescent children in the highest grades of primary school value PAL. When developing PAL programs for the classroom setting, it is therefore relevant to consider children’s enjoyment of the program in addition to the effects of the program regarding academic outcomes.

In the current study, we assessed the effects of a 5-week program that integrated juggling with math for 10- to 11-year-old primary school children on math memorization performance measured using a multiplication test. The intervention group combined a new, challenging motor task (juggling) with academic content, while the non-active control group received the same academic content as the intervention group, but without PA. Additionally, we assessed children’s enjoyment of the intervention and control lessons.

## 2. Materials and Methods

### 2.1. Recruitment of Participants

We invited a convenience sample of approximately 150 primary schools from the network of the researchers using email to participate in our study. In addition, we spread the invitation via social media channels (i.e., Facebook and LinkedIn). Forty-one schools across the Netherlands showed interest in response to our invitation. We provided them with further information about the study’s aim, procedures, and requirements via email and/or telephone, and asked them if they were willing to participate. One school did not respond and five schools withdrew due to a lack of time or inability to adjust their schedule to the study requirements. Twenty-five schools that had signed up were excluded because they did not meet the inclusion criteria, i.e., solely fifth grade students (N = 10); more than twenty children per class (N = 3); two or more fifth grade classes in the same school or one fifth grade class with a school nearby with similar environmental conditions that had also signed up (N = 12). In addition, one school was excluded since they were not willing to provide us with data on the children’s academic performance. After a two-week recruitment period and a two-week selection period, we included nine schools that were geographically spread across the Netherlands. 

Prior to the start of the study, we informed all fifth grade children (N = 369) and their parents about the study’s aims and procedures by means of an information letter. We agreed with the schools that both the intervention and control math lesson program were to be implemented as part of the daily math curriculum. Only children with written permission of their parent/caregiver were allowed to participate in the measurements. The study was approved by the Medical Ethical Committee of the VU University Medical Centre Amsterdam [2014.363].

### 2.2. Study Design and Randomization

We conducted a cluster-randomized controlled trial. Before the start of the experiment, we randomly assigned the participating classes (N = 14) to the intervention or control group by asking the teacher to draw an envelope that included a lot that said either “intervention” or “control.” The randomization was stratified by school; in schools with two fifth grades, we ensured that if one class was assigned to the control group, the other class was assigned to the intervention group. We applied the same procedure for schools with one grade 5, i.e., when one school drew the lot of the intervention group, the nearby school with similar environmental conditions was assigned to the control group. As such, we aimed to include children with similar background characteristics in the control and intervention groups.

### 2.3. Experimental Procedure

The research team visited all schools three times between March and May 2017. The first visit was scheduled prior to the start of the experiment to introduce the teachers to the research procedures and the intervention or control program. The second visit took place in the week prior to the start of the intervention or control program. During this visit, the researcher (A.K.) provided children with detailed instructions about the experiment and conducted the pretest measurements. We asked all children with informed consent to fill out three short questionnaires for descriptive purposes (see section “measurement instruments”). To assess math memorization performance, we conducted a multiplication table test with the entire class. We measured children’s juggling performance in a separate room in groups of 3–6 children. The week after the pretest measurements, each class started with the 5-week intervention or control program. Each week, we sent an email to all teachers with an update for the next week and a brief evaluation on implementation of the program. The week after the final intervention or control lesson, we visited the schools for the third time to conduct the post-intervention measurements. This post-test included a multiplication tables test identical to the pretest. Furthermore, children filled out a short evaluation questionnaire.

### 2.4. Intervention and Control Programs

The intervention and control programs consisted of the same math lessons, developed by experts in educational arithmetic methods and focused on the memorization of multiplication tables 2 to 9. The 5-week program consisted of 20 math lessons (four lessons per week), with a weekly increasing of the difficulty of the multiplication tables. The tables were practiced in different forms, e.g., repeating the multiplication tables in forward order, backward order, by letting the girls and boys answer alternately, or by skipping one of the multiplication sums in a line (e.g., 2 × 6, 4 × 6, 6 × 6, etc.). Sums practiced in the intervention and control groups were exactly the same, with the only difference between the two groups being the addition of juggling exercises to the math lessons in the intervention group.

#### 2.4.1. Control Program

The control lessons lasted approximately 5 min, wherein the multiplication tables were practiced collectively under the supervision of the classroom teacher. The teacher received a manual with the instructions and sums for each lesson. The teacher read out loud the multiplication tables sums and children were asked to answer the sums verbally while sitting behind their desk.

#### 2.4.2. Intervention Program

Experienced educators, each with a background as a physical education teacher, developed and produced 20 instruction videos in which a juggler shortly introduced the juggling exercises and then combined the juggling exercises with the multiplication tables. Each lesson lasted approximately 5 to 8 min. The complexity of the juggling exercises increased each week, i.e., practicing with easy throw and catch exercises with one ball in week 1 and 2, two balls in week 3 and 4, and ending with using three balls in week 5 of the program. Children were instructed to answer the multiplication table sums given by the juggler verbally when they caught the ball. A fixed rhythm was used for throwing and catching the balls and answering the math sums. Classroom teachers received a password to access the videos of the intervention program via a secured part of a webpage (www.smart-moves.nl). The videos were displayed on digital screens in the classrooms while children were standing and juggling behind their desks. Teachers also received an instruction manual including an overview of the lessons and tips (e.g., let children stand behind their desk, such that a missed ball will land on the table instead of the floor).

### 2.5. Measurement Instruments

#### 2.5.1. Descriptive Characteristics

At baseline, children self-reported their birth date and gender. The physical activity questionnaire for children (PAQ-C) was administered to measure children’s moderate-to-vigorous physical activity (MVPA) during a regular week. The questionnaire covered physical activity during and after school time [39,40]. A summary score of 1 on the PAQ-C indicates low physical activity, while a score of 5 indicates a high activity level [39]. The PAQ-C has been shown to have a high internal consistency (alpha = 0.79), moderate convergent and construct validity (r = 0.63 and r = 0.57, no differences between boys and girls), and test–retest reliability (r = 0.75 for girls, r = 0.82 for boys) [39,40]. The subscale “scholastic skills” of the Dutch version of the Harter’s Self Perception Profile for Children (in Dutch: Competentie Belevingsschaal voor Kinderen; [41]) was used to measure children’s perceived scholastic competence. The questionnaire, including the subscale called scholastic competence, has sufficient construct validity and good test–retest reliability (Intraclass Correlation Coefficient (ICC) = 0.85) in Dutch children [42,43].

We asked schools to provide standardized test scores of the participating children in reading comprehension, orthography, and math/arithmetic obtained from the Dutch norm-referenced CITO test battery [44]. In many Dutch schools, the CITO test battery is administered twice a year to track children’s performance levels. Furthermore, teachers that provided the physical education (PE) classes (either classroom teachers or PE teachers) were asked to rate children’s level of motor skills in the regular physical education lessons using the motor skill levels as defined in a didactic handbook for physical education teachers in the Netherlands (in Dutch “Het Basisdocument”) [45]. Teachers categorized children according to a four-point scale (see Table 1).

We assessed children’s juggling performance using juggling exercises based on the steps for juggle didactics in the same handbook (in Dutch “Het Basisdocument”; [45]). A description of the test can be found in Table 2. Before the start of the juggling test, we asked each student to self-report his/her hand of preference. The juggling test consisted of eight exercises with increasing difficulty. Before each exercise, the researcher provided the children with instructions and an example. The children in the group were asked to simultaneously perform each exercise ten times on a pace verbally set by the researcher. Two research assistants observed the children during the test and scored their performance for every throw (i.e., dichotomous assessment: catch or no catch). For each exercise, children’s scores ranged between 0 (ten balls missed) and 10 (ten balls caught). In case a child had five or more errors, the test ended for this child and he/she automatically received a score of “0” for the next level. Children who had five or more errors were offered to continue juggling or to encourage their classmates. When all children had five or more errors, the test was ended.

#### 2.5.2. Primary Outcome: Multiplication Memorization Performance

A multiplication tables test was developed together with the same experts who developed the math lesson program. The test measured children’s knowledge and skills as practiced during the math lesson program (i.e., memorization of multiplication tables). Prior to the test, the researcher provided children with instructions about the test and an example sum. During the test, the researcher read each multiplication sum to the class out loud. Children were asked to answer the questions individually by writing down the answer on their individual test sheet. The test consisted of 30 random multiplication sums, using the same tables (2 to 9) and speed (i.e., 2 s) as in the math lesson programs. The pre- and post-test were identical, and took approximately 4 min. For the analyses, we used the number of correct answers, ranging from 0 to 30, as a dependent variable.

#### 2.5.3. Secondary Outcome: Enjoyment

During the posttest, we asked children to fill out a short evaluation questionnaire in which they rated their perceived enjoyment during the math lesson program (control or intervention program) on a 10 cm visual analog scale, ranging from “Not at all” to “Very much.” Scores for the variable “enjoyment” ranged from 0 to 10. Furthermore, we asked children if they would like to participate in a similar math lesson program again (i.e., the juggling-math for the intervention group or math lessons for the control group). Answers were scored on a three-point scale with smiley icons representing the answer options: happy smiley = “Yes, please” (2); neutral smiley = “Don’t know/no opinion” (1); or sad smiley = “No, rather not” (0).

#### 2.5.4. Intervention Integrity

After the baseline measurements, we provided each class with a poster on which the children and teacher marked each finished intervention or control lesson by means of a sticker. After the experiment, we calculated the proportion of implemented control/intervention lessons per class (i.e., 100% equals 20 lessons).

#### 2.5.5. Data Analyses

We analyzed all data using the Statistical Package for Social Sciences (IBM SPSS statistics 23, IBM Corp, Armonk, NY, USA). We used descriptive statistics to report baseline scores. The effect of the intervention on multiplication performance was tested using a mixed-model analysis, including group (control or intervention) as a fixed factor and class and school as random intercepts. Pretest multiplication performance score, age, gender, motor skill level, PAQ-C score, and CITO math/arithmetic score were included as covariates. We also analyzed differences in enjoyment using mixed-model analysis, including group as a fixed factor, and class and school as random intercepts. Pretest multiplication performance, age, gender, motor skill level, PAQ-C score, and CITO math/arithmetic score were included as covariates. The level of significance was set at *p* < 0.05. We applied the intention-to-treat principle, meaning that all children with complete data (multiplication performance, enjoyment, and all covariates) were included in the data analyses.

## 3. Results

### 3.1. Study Sample and Descriptive Characteristics

A total of 323 (88%) children from nine elementary schools returned their informed consent form and were included in the study (see Table 3). We analyzed data from 299 children who had complete and valid data on the outcome measures and all covariates. A common reason for not being included in the data analyses was absence during one of the measurements. Baseline scores of children in the intervention and control group were similar, except for the multiplication tables test that was slightly higher in the intervention group (mean: 24.3 (SD = 5.9)) than in the control group (mean: 21.4 (SD = 7.2)).

### 3.2. Intervention Effects

We found no significant intervention effect on multiplication memorization performance (see Table 4). However, children in the intervention group reported to have enjoyed the lessons significantly more than children who followed the sedentary math group (see Table 4). In the intervention group, 70% of the children reported that they would like to follow the intervention program (i.e., combined juggling-math lessons) more often. In the control group, 20% of the children indicated that they would like to follow the sedentary math lesson program more often.

### 3.3. Intervention Integrity

Teachers in the intervention classes implemented, on average, 98% of the juggling-math lessons. In the control classes, an average of 99% of the math lessons was implemented during the 5-week experimental period.

## 4. Discussion

A 5-week math program integrated with juggling exercises did not significantly improve, nor deteriorate, 10- to 11-year-old children’s multiplication memorization performance as compared to children that followed the same math program without juggling. Children enjoyed the combined juggling-math program significantly more than the sedentary math program. The high degree of implementation indicates that the PAL program was feasible to implement in grade 5 of primary school.

Our results are in line with studies of Riley et al. [18] and Donnelly et al. [17] who found no improvements in math performance after short (6 weeks) or long (3 years) PAL programs, respectively, compared with a non-active control group. Donnelly et al. implemented a 3-year program including daily 2 × 10 min PAL lessons of moderate-to-vigorous intensity for children that were initially 8 years old [17], while Riley et al. implemented a 6-week moderate-to-vigorous intensity PAL program in which lessons were implemented 3 times a week for 60 min in 10- to 12-year-old children [18]. A recent study of Fedewa et al. reported that relatively short 10-min, academic-based PA breaks (e.g., dancing while answering questions that appear on the digital screen) did not improve children’s math performance in grade 3 and 4 after nine months, compared to 10 min PA breaks not integrated with academic content [23]. However, the lack of a no-PA control group in the before-mentioned study makes it difficult to interpret the effects reported. Although there are several differences between above-mentioned studies and our study, such as the duration and intensity of the PAL (moderate-to-vigorous intensity versus light intensity in our study), duration of the intervention period, degree of integration of PA with academic content, and age group, none of the studies showed beneficial nor adverse effects of PAL on children’s math performance.

In contrast, several other studies reported improved math performance after PAL programs [8,9,10,12,15,16,22]. Differences in results can be due to several factors. First, the majority of previous studies reporting positive effects included younger children, i.e., 4 to 9 years old versus 10 to 11 years old in our study. Only the studies of Vazou and Skrade (4th and 5th grade = 9–11 years old) and Fedewa et al. (8–11 years old) included children of similar age. In contrast to our study, these studies showed positive effects of PAL on math performance [10,15]. Due to inconsistencies in the outcomes of the studies in preadolescent children, it is currently not possible to draw conclusions with regard to the effectiveness of PAL for this age group. Second, studies are difficult to compare due to differences in the difficulty levels of the PAL programs. The studies reporting positive effects mainly combined simple, easy-to-perform PA with academic tasks. In contrast, in our study, the children participated in a more complex motor task, namely learning to juggle. It could be that (some of) the juggling exercises were too difficult for the children, resulting in a too high cognitive load, hindering improvements in math performance. A third reason for differences in findings may lie in the duration of the PAL lessons and the intervention period. We developed a relatively small PAL program, with short lessons (5 to 8 min) over a short duration (5 weeks). This is in contrast to several earlier studies that implemented PAL over longer durations (>10 min) and measured children’s performance after multiple months or years (e.g., see References [8,10,16,22]). A study with comparable intervention duration (6 weeks) implemented considerably longer PAL lessons, i.e., 3 × 60 min per week [9]. Therefore, it could be that our PAL program was too short to affect math memorization performance. Future research is needed to gain insight into the effects of duration and difficulty to create effective PAL. Researchers are also recommended to examine whether effects depend on age and differ for children with varying levels of academic performance and/or motor skills. Fourth, differences in outcomes may be due to the intensity of the PAL programs. Most studies reporting positive effects implemented PAL of moderate-to-vigorous intensity, while the lessons in our PAL program were of light intensity. The exact role of the PA intensity in the effectiveness of PAL programs is still unclear and needs to be further examined [7]. Fifth, it could be that children benefitted from the PAL program, but that differences in performance are only measurable over time. Therefore, we recommend future research to include longer-term follow-up measurements to assess delayed intervention effects. A last reason for not finding positive effects on math memorization performance may be found in our measurement instrument. It appeared that the children had already high baseline scores on the multiplication table test, which may have caused limited room for improvements.

In line with the studies of Mavilidi and colleagues [12,13,14], we found that children enjoyed the PAL significantly more than the sedentary version of the math lessons. In addition, the vast majority of children were enthusiastic about following the PAL more often. Thus, PAL is not only enjoyable for young children, who by nature explore and learn by moving around [46,47], but also for older children in higher grades of primary school. This is an important finding given the fact that some teachers question the use of PAL for this age group [34,36]. Moreover, PAL can be a good solution to meet children’s need to alternate long periods of sitting during the school day [30], without “losing” time for academic tasks. It has also been suggested that the increased enjoyment/positive affect caused by PA can subsequently result in higher school engagement and improved academic performance [38]. This is particular relevant to the age group of (pre)adolescent children, in which school engagement tends to decline [38]. Further research is needed to examine whether our juggling-math lessons can improve math performance over the longer-term, with enjoyment and school engagement as potential mediators. When PAL is implemented over longer periods of time, it is important to provide a sufficient challenge and variety in the lesson program.

Implementation of the juggling-math lessons was excellent (on average 98% of the lessons), and thus the lessons seem feasible in daily school practice. This is very promising since most school-based PA programs reported disappointingly low implementation rates [32]. As the aforementioned programs mostly focused on non-integrated PA, it could be the case that the integration of academic content with PA is critical for successful implementation of a PA program in the classroom by primary school teachers. This is in line with studies reporting on the perceptions of principals and teachers that PA must be supportive of academic tasks [34,48,49]. Besides, it could be that implementation rates were high because the PAL lessons were digitally provided, which means that teachers were not required to prepare and execute the lessons themselves. We must note that teachers were provided with reminders during the intervention period to stimulate implementation of the PAL. Further research is therefore needed to gain insight in implementation outside the research setting, and over longer periods. Herein, gaining more insight in teacher-perceived factors that facilitate implementation is recommended (e.g., integration of PA with academic content, duration of the lessons/program).

### Strengths and Limitations

Our study had several strengths, such as the randomized controlled trial design and high intervention integrity. Furthermore, the juggling-math program was developed by educational experts with substantial knowledge and experience in physical education and math education. Therefore, the PAL was connected to the Dutch educational goals and fitted well in the existing school curricula. Lastly, the intervention and control program consisted of the same math lessons, which is a major strength compared to earlier studies that contrasted PAL with the regular curriculum consisting of other academic content. Some limitations should also be considered. First, we were not able to blind the outcome assessors. Second, we had no baseline measures of math enjoyment. Third, the multiplication test was quite easy for the children, causing ceiling effects on the post-test. Fourth, due to the juggling instructions, the juggling-math lessons were slightly longer (5 to 8 min) compared to the math lessons in the control group (5 min). Lastly, although implementation rates were excellent, we have no insight into how well individual children participated in the juggling-math program (i.e., if the juggling exercises were too difficult or not, and in which stage of the 5-week program).

## 5. Conclusions

Five-week integrated juggling-math lessons did not improve, nor deteriorate, 10- to 11-year-old children’s math performance. Children enjoyed the juggling-math lessons significantly more than the sedentary math lessons. The lessons appeared feasible to implement in the daily school curriculum. The beneficial effects on enjoyment may motivate teachers and schools to implement PAL in their classrooms in future.

## Figures and Tables

**Table 1 ijerph-16-02452-t001:** Description of motor skill levels according to Mooij et al. [45].

Level	Description
**0**	For this student, the PE lesson offered is usually too difficult and fails the activity regularly (or only succeeds with the help of the teacher).
**1**	For this student, the PE lesson offered is appropriate and the activity usually succeeds. The execution is not yet efficient or smooth.
**2**	Similar to level 1, but the execution is efficient and smooth.
**3**	For this student, the PE lesson offered is often too easy. Execution is immediately smooth and efficient, and the student is looking for new challenges.

**Table 2 ijerph-16-02452-t002:** Description of the juggling test.

	Exercise	Description
**0**	Throw and catch with one ball	Throw the ball vertically with one or two hands and catch it with both hands.
**1**	Pillar with one ball	Throw one ball vertically and catch the ball with the same hand. First with the hand of preference and then with the other hand.
**2**	Crossing with one ball	Throw the ball from one hand to the other with a bow. First with the hand of preference and then with the other hand.
**3**	Pillar with two balls	In each hand one ball and throw them vertically. Catch the balls with the same hand as thrown.
**4**	Crossing with two balls	In each hand one ball and throw them with a bow to the other hand. The second ball is thrown when the first is at the peak of its bow.
**5**	(Cascade) Juggling with three balls	Juggle with three balls.

**Table 3 ijerph-16-02452-t003:** Baseline characteristics (means and standard deviations *).

Baseline Characteristics	Total Group (N = 323)	Intervention Group (N = 170)	Control Group (N = 153)
Age (years, mean)	11.0 (0.45)	11.0 (0.42)	10.9 (0.48)
Gender (%, boys/girls)	52/48	50/50	45/55
Motor skill level (n (%))	N = 306	N = 170	N = 136
0	8 (2.6)	5 (2.9)	3 (2.2)
1	71 (23.2)	42 (24.7)	29 (21.3)
2	161 (52.6)	80 (47.1)	81 (59.6)
3	66 (21.6)	43 (25.3)	23 (16.9)
PAQ-C	N = 312 3.3 (0.82)	N = 165 3.3 (0.73)	N = 147 3.4 (0.91)
CBSK	N = 254	N = 130	N = 124
scholastic competence	17.3 (3.6)	17.6 (3.5)	16.9 (3.7)
CITO	N = 320/318/313	N = 168/168/165	N = 152/150/148
reading comprehension	48.6 (12.9)	49.3 (12.2)	47.8 (13.5)
orthography	141.8 (6.7)	142.2 (6.3)	141.3 (7.2)
math/arithmetic	103.5 (12.0)	104.1 (12.2)	102.8 (11.6)
Juggling performance (median)	N = 308	N = 163	N = 145
exercise 0	10	10	10
exercise 1a	10	10	10
exercise 1b	10	10	10
exercise 2a	10	10	10
exercise 2b	10	10	10
exercise 3	7	7	7
exercise 4	2	2	2
exercise 5	0	0	0

* Unless stated otherwise; PAQ-C: physical activity questionnaire for children; CBSK (Competentie Belevingsschaal voor kinderen): Dutch version of the Harter’s Self Perception Profile for Children.

**Table 4 ijerph-16-02452-t004:** Post-intervention multiplication performance, enjoyment, and intervention effect.

Outcome Measures	Intervention Group (Mean + SD) (N = 163)	Control Group (Mean + SD) (N = 136)	Intervention Effect (Beta, 95% Confidence Interval *)
Multiplication performance	25.9 (4.8)	23.3 (7.1)	0.4 [−0.4; 1.2]
Enjoyment	7.0 (2.5)	4.7 (2.3)	−2.2 [−3.2; −1.1]

* Multilevel regression analyses adjusted for pretest multiplication performance score, age, gender, motor skill level, CITO math/arithmetic score, and PAQ-C score.

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
