# Peer review of "Integrating Juggling with Math Lessons: A Randomized Controlled Trial Assessing Effects of Physically Active Learning on Maths Performance and Enjoyment in Primary School Children"

_ijerph, 2019, doi:10.3390/ijerph16142452_

Round 1

Reviewer 1 Report

I would like to congratulate the authors on a nicely written, well-designed, thorough report of their study, it was a pleasure to read. In particular, the introduction and discussion convey a strong understanding of research in this area and raise several important points for others doing research on this topic to consider. I don't see the need for any substantive edits, I have a few minor suggestions which may aid clarity:

Introduction, lines 71-72: To assist a reader's understanding, perhaps the authors could provide examples of 'cognitively engaging' and 'non cognitively engaging' physical activities.

Introduction, line 99: I respectfully suggest the word 'deserved' is replaced with 'received', for clarity.

Methods, line 233: I respectfully suggest the phrase 'in case' is replaced with 'when', for clarity.

Results, Table 3: In the cells containing information on motor skill level for the intervention and control group participants, the %'s in the intervention group add up to 97%, and the %'s in the control group cell add up to 89% - should the %'s in each of those cells add up to 100%? 

Author Response

Response to reviewer 1

I would like to congratulate the authors on a nicely written, well-designed, thorough report of their study, it was a pleasure to read. In particular, the introduction and discussion convey a strong understanding of research in this area and raise several important points for others doing research on this topic to consider. I don't see the need for any substantive edits, I have a few minor suggestions which may aid clarity.

We thank the reviewer for his/her kind words on our manuscript and his/her suggestions for improving/clarifying some part in our manuscript. We considered all the suggestions and made adaptations to our manuscript accordingly (changes highlighted in yellow). Below we address the reviewer’s suggestions point by point.

Suggestion 1:

Introduction, lines 71-72: To assist a reader's understanding, perhaps the authors could provide examples of 'cognitively engaging' and 'non cognitively engaging' physical activities.

The reviewer suggests to add some examples of cognitively engaging and non-cognitively engaging physical activities. We agree examples improve clarity and added them in.

Line 71-72: Recent meta-analyses reported larger effects of long-term cognitively engaging PA (e.g. coordinative PA, team games or complex motor skill training that require cognitive effort) on academic performance compared to non-cognitively engaging PA (e.g. simple, repetitive aerobic exercises) in children [3,5].

Suggestion 2:

Introduction, line 99: I respectfully suggest the word 'deserved' is replaced with 'received', for clarity.

We thank the reviewer for this suggestion and adjusted the sentence accordingly. The new sentence reads as follows:

Line 99: Another important aspect of PAL that received relatively little attention so far is the effect that PAL may have on children’s enjoyment.

Suggestion 3:

Methods, line 233: I respectfully suggest the phrase 'in case' is replaced with 'when', for clarity.

We thank the reviewer for this suggestion and replaced ‘in case’ with ‘when’. The new sentence reads as follows:

Line 233: When all children had 5 or more errors, the test was ended.

Suggestion 4:

Results, Table 3: In the cells containing information on motor skill level for the intervention and control group participants, the %'s in the intervention group add up to 97%, and the %'s in the control group cell add up to 89% - should the %'s in each of those cells add up to 100%? 

We thank the reviewer for noticing this mistake in Table 3. We replaced the incorrect percentages with the correct ones. The percentages now all add up to 100% (Total group 2.6+23.2+52.6+21.6 = 100; Intervention group 2.9+24.7+47.1+25.3 = 100; Control group 2.2+21.3+59.6+16.9 = 100).

Reviewer 2 Report

Dear Authors,

I'd like to congratulate of very informative and smoothly flowing text with excellent methodology - its' quality and description. 

These few comments might only be a voice in discussion around your work, and for the future:

- little is known about the effects of physically active learning (line 22), but a lot about an active learning with the brain hemispheres integrartion by movement.

- the lack of significant differences in childrens performance during this study does not prejudge about them, ie. children from experimental group might remember longer or stronger than controll ones, the result might need time. 

- positive feelings and attitudes about learning are the foundation of school success and cognitive development, so your results are really nice!

- enjoynment from ie. active math classes could possibly be a way to improve daily physical activity, and create a healthy need/habit for life - imagine breaks at work with juggling - so beneficial.

I wish you all the best.

Author Response

Response to reviewer 2

Dear Authors,

I'd like to congratulate of very informative and smoothly flowing text with excellent methodology - its' quality and description. 

I wish you all the best.

We thank the reviewer for his/her kind words on our manuscript and his/her suggestions and comments. We provide a point-by-point response to the reviewer’s comments below.

These few comments might only be a voice in discussion around your work, and for the future:

1.- little is known about the effects of physically active learning (line 22), but a lot about an active learning with the brain hemispheres integration by movement.

The reviewer raises an interesting point which is beyond the scope of the current study.  We agree this may be voiced in future studies

2. - the lack of significant differences in children’s’ performance during this study does not prejudge about them, ie. children from experimental group might remember longer or stronger than control ones, the result might need time. 

We thank the reviewer for this comment. We have added a sentence to our discussion section to highlight the possibility that effects may occur after longer time periods and added a recommendation for future research.

Line 353-355: Fifth, it could be that children benefitted from the PAL program, but that differences in performance are only measurable over time. Therefore, we recommend future research to include longer-term follow-up measurements to assess delayed intervention effects.

3. - positive feelings and attitudes about learning are the foundation of school success and cognitive development, so your results are really nice!

4. - enjoyment from ie. active math classes could possibly be a way to improve daily physical activity, and create a healthy need/habit for life - imagine breaks at work with juggling - so beneficial.

We agree with the reviewer on the two above-mentioned conclusions (3 and 4). We hope that physically active lessons will be used more frequently in schools both to foster school enjoyment and provide additional opportunities for children to be physically active.